# Dietary High Glycinin Reduces Growth Performance and Impairs Liver and Intestinal Health Status of Orange-Spotted Grouper (*Epinephelus coioides*)

**DOI:** 10.3390/ani13162605

**Published:** 2023-08-12

**Authors:** Yanxia Yin, Xingqiao Zhao, Lulu Yang, Kun Wang, Yunzhang Sun, Jidan Ye

**Affiliations:** 1Xiamen Key Laboratory for Feed Quality Testing and Safety Evaluation, Fisheries College, Jimei University, Xiamen 361021, China; 202112951023@jmu.edu.cn (Y.Y.); 18328817044@sina.cn (X.Z.); 202112951038@jmu.edu.cn (L.Y.); wangkun@jmu.edu.cn (K.W.); jmusunyunzhang@163.com (Y.S.); 2Fujian Engineering Research Center of Aquatic Breeding and Healthy Aquaculture, Fisheries College, Jimei University, Xiamen 361021, China

**Keywords:** glycinin, inflammatory reaction, intestinal flora, apoptosis, *Epinephelus coioides*

## Abstract

**Simple Summary:**

Soy antigen proteins are regarded as the cause of fish enteritis, of which glycinin is one of the main soy antigen proteins. By comparing different levels of glycinin within a high-soybean-meal diet, we found that dietary high glycinin (10%) could decrease hepatic antioxidant ability, lead to microbiota dysbiosis, and change the relative abundance of some intestinal microbiota and intestinal morphology, thereby causing intestinal inflammation. The results showed that dietary high glycinin (10%) may be the potential trigger of soybean-induced enteritis of orange-spotted grouper (*Epinephelus coioides*).

**Abstract:**

The aim of the study was to investigate whether the negative effects of dietary glycinin are linked to the structural integrity damage, apoptosis promotion and microbiota alteration in the intestine of orange-spotted grouper (*Epinephelus coioides*). The basal diet (FM diet) was formulated to contain 48% protein and 11% lipid. Fish meal was replaced by soybean meal (SBM) in FM diets to prepare the SBM diet. Two experimental diets were prepared, containing 4.5% and 10% glycinin in the FM diets (G-4.5 and G-10, respectively). Triplicate groups of 20 fish in each tank (initial weight: 8.01 ± 0.10 g) were fed the four diets across an 8 week growth trial period. Fish fed SBM diets had reduced growth rate, hepatosomatic index, liver total antioxidant capacity and GSH-Px activity, but elevated liver MDA content vs. FM diets. The G-4.5 exhibited maximum growth and the G-10 exhibited a comparable growth with that of the FM diet group. The SBM and G-10 diets down-regulated intestinal tight junction function genes (*occludin*, *claudin-3* and *ZO-1*) and intestinal apoptosis genes (*caspase-3*, *caspase-8*, *caspase-9*, *bcl-2* and *bcl-xL*), but elevated blood diamine oxidase activity, D-lactic acid and endotoxin contents related to intestinal mucosal permeability, as well as the number of intestinal apoptosis vs FM diets. The intestinal abundance of phylum Proteobacteria and genus *Vibrio* in SBM diets were higher than those in groups receiving other diets. As for the expression of intestinal inflammatory factor genes, in SBM and G-10 diets vs. FM diets, pro-inflammatory genes (*TNF*-α, *IL-1β* and *IL-8*) were up-regulated, but anti-inflammatory genes (*TGF-β1* and *IL*-*10*) were down-regulated. The results indicate that dietary 10% glycinin rather than 4.5% glycinin could decrease hepatic antioxidant ability and destroy both the intestinal microbiota profile and morphological integrity through disrupting the tight junction structure of the intestine, increasing intestinal mucosal permeability and apoptosis. These results further trigger intestinal inflammatory reactions and even enteritis, ultimately leading to the poor growth of fish.

## 1. Introduction

World aquaculture currently provides most of the world’s food fish consumption, demonstrating its crucial role in global food security [1]. However, fish production, especially intensive fish farming, depends heavily on the supply of artificial feed [2]. Furthermore, marine fish requires more and a higher quality of feed protein than terrestrial animals do. Processed products of marine catches such as fish meal (FM) are the preferred animal protein source in the feed formulations of marine fish due to their high nutritional value [2]. Given the global constrained supply of FM, reducing the dependence of aquafeeds on FM is widely accepted as an important strategy for ensuring the developments of aquaculture in the future [3].

One of the major sources of alternatives to FM is soybean by-products, and they have been widely used in aquafeeds as the main protein for most farmed fish because soybean has a higher protein quality, steady supply, and low price compared with other vegetable proteins [4]. It has been reported that the use of a high-soybean diet can cause poor growth performance [5,6,7], induce oxidative stress [8,9], increase histological damage in the intestine [10,11], lead to microbiota dysbiosis [7,8,9] and induce intestinal inflammation [12,13,14]. However, the occurrence of the negative effects of SBM is caused by the existence of anti-nutritional factors (ANFs) [15,16]. Soy antigen proteins (SAPs), a form of globulin, are one of the main ANFs in soybean by-products [17,18]. This protein has high thermal stability [19] and strong immunogenicity [20], and it is difficult to eliminate by conventional treatment methods [21]. This kind of protein is believed to be the cause of enteritis of both terrestrial animals and fish [21]. SAPs can be subdivided into four subtypes, including glycinin, α-conglycinin, β-conglycinin and γ-conglycinin, by the sedimentation rates of the proteins, of which glycinin and β-conglycinin account for about 40% and 31% of total soy globulin, respectively. Therefore, the majority of the research focuses on the sensitization of the two types of SAPs in animals and their underlying mechanisms. Recent results showed that SAPs exhibit high antigenic activity [22,23,24,25,26], induce specific antigen–antibody reactions and delay hypersensitivity mediated by T lymphoid cells—by stimulating the immune system [15,22,27,28]; trigger reactive oxygen, leading to oxidative stress in intestinal epithelial cells [5,27,29,30] and intestinal epithelial cell apoptosis and damage [15,28,31,32]; and develop into symptoms such as enteritis and allergic reactions [22,24,26,33], eventually leading to poor growth [34]. In terms of oxidative stress, many factors in diets, such as polypropylene microplastics [35] and nitrite [36], can cause oxidative stress in fish and injure tissues and organs. Fish, especially farmed marine fish, have lower tolerance to SAPs than do terrestrial animals, which is evidenced by a number of experimental results in FM replacement by an excessive dietary addition of soybean by-products [4,34]. Although SAPs have a negative impact on fish growth and health, the underlying exact mechanism involving enteritis and allergy induced by SAPs is still unclear. In the future, more relevant research work needs to be carried out in order to find better feeding strategies and feed formulations to ultimately improve the health and growth of fish.

Grouper aquaculture has developed into a large-scale aquaculture industry along the coast of Southeast Asia [37], with the largest aquaculture production scale in China. China’s output of farming grouper reached 204,119 tons in 2021 [38], ranking it as the third-largest mariculture fish in China. The grouper is highly favored by Chinese people, not only because of their traditional consumption habits, but also because of its rapid growth and high-quality flesh. Similar to other carnivorous fish, grouper is also highly sensitive to soybean by-products [4,10,39]. In these studies, feeding high soybean by-products caused the decline of both the feed intake and the growth of the grouper, accompanied by obvious symptoms of intestinal mucosal structure damage. Recently, the effects of dietary glycinin and β-conglycinin on the immunity and microbiota in the intestine of hybrid grouper (*Epinephelus fuscoguttatus* ♀ × *Epinephelus lanceolatus* ♂) were investigated [40]. However, the effects of SAPs on apoptosis and permeability in the intestine of grouper have not been investigated yet. In this study, we sought to investigate the effects of dietary glycinin levels on the growth, intestinal barrier function, intestinal cell apoptosis and intestinal microbiotia in the intestine of orange-spotted grouper (*Epinephelus coioides*). With these results, the relationship between intestinal health and SAPs in the fish species is better understood and its mechanism of action is clarified, which provides a clue to reasonably develop and utilize soybean by-products for grouper feed. This study provides a new basis for the prevention of SAP-induced physiological responses in this fish species.

## 2. Materials and Methods

### 2.1. Experimental Diets

A FM diet containing 48% crude protein and 11% crude lipid was prepared, using FM as the protein source. The 30% FM (60% FM protein) was replaced by SBM in the FM diet to make a high-SBM diet. In order to determine the effects of SAPs in SBM, 4.5% and 10% glycinin (79.97%, provided by Professor Guo Shuntang from China Agricultural University) were added in the FM diets to make a G-4.5 diet and a G-10 diet. The dietary 4.5% and 10% glycinin used were equivalent to the substitution of SBM for 30% and 60% FM protein, respectively. The dietary glycinin levels were adjusted in parallel with decreases in the casein and gelatin levels in order to keep the crude protein and lipid levels of G-4.5 and G-10 diets identical to those in the FM and SBM diets. All raw feed materials were crushed and sifted through a 60-mesh sieve, then weighed and homogenized. Then, an appropriate amount of water was added to make granular feed with a diameter of 2.5 mm. Feeds were placed in an oven at 55 °C for 12 h, sealed with a bag, and stored in a refrigerator at −20 °C until feeding. The composition and nutritional level of the test feed are given in Table 1.

### 2.2. Feeding Trial

The growth trial was carried out at Fujian Dabeinong Fisheries Technology Company (Zhangzhou, China). Before the beginning of the trial, the fish were fed with commercial grouper feed (Jiakang Feed Co. Ltd., Xiamen, China; crude protein, 49.3%, crude lipid, 12.7%) for a 4-week acclimatization. A total of 240 orange-spotted grouper (*Epinephelus coioides*) of a similar size (initial body weight: 8.01 ± 0.10 g) were randomly distributed into four test groups with triplicate tanks (450-L) within a recirculating rearing system with a water temperature control device, at a stock density of 20 fish per tank and a flow rate of 8 L/min of seawater per tank. Fish in each group were hand-fed one of the diets to apparent satiation at each meal twice daily (7:30 and 17:30) during an 8-week growth trial period. The remaining diets were gathered and then feces were removed by siphoning 30 min after each meal. The collected diets were dried and weighed to determine the feeding rate (FR). Due to the seawater loss of the rearing system caused by daily cleaning, fresh seawater was filled to the original level of the pool water. During the period of growth trial, the water temperature fluctuated between 28–29 °C, the ammonia nitrogen concentration was below 0.2 mg/L, and the dissolved oxygen was higher than 6.60 mg/L.

### 2.3. Sample Collection

At the termination of the 8-week growth trial, the fish were collected by tank, anesthetized with MS-222 solution (100 mg/L), and bulk-weighed and calculated to determine percent weight gain (WG), specific growth rate (SGR), and the feed efficiency (FE). Thirty fish in each group (ten fish per tank) were randomly seized and weighed individually, after anesthesia with MS-222 solution, to calculate the hepatosomatic index (HSI) and condition factor (CF). Blood was collected from the caudal veins of ten fish per tank using a sterile 2-mL heparinized syringe, then transferred to 1.5-mL tubes. After centrifugation of blood samples (1027× *g*, 10 min, 4 °C), the plasma samples were pooled by tank and stored at −80 °C until the analysis of biochemical components was performed. After completing the blood collection, the fish were killed by a sharp blow on the head and the abdominal cavity was dissected out under aseptic conditions, followed by intestinal removal. The intestine was opened with a sterile scalpel and the feed digesta was washed off with saline solution, according to the method in our previous studies [42]. The intestine was pooled by tank (nine fish per replicate tank) and stored at −80 °C until the determination of digestive enzyme activity, permeability and gene mRNA expression of inflammatory cytokines in the intestine was to be carried out. The intestines of the remaining fish were dissected into proximal, mid and distal parts (PI, MI and DI respectively) as described [43]. The three segments were then fixed in 4% formalin PBS for histological analysis and terminal deoxynucleotidyl transferase-mediated dUTP nick-end labeling (TUNEL) assay. Another four animals from each tank were randomly caught, placed into plastic bags, and stored at −20 °C before the determination of whole-body proximate composition was carried out.

### 2.4. Proximate Composition Determination

The proximate composition of feed ingredients, diets, and whole-body fish samples was determined according to the method of the Association of Official Analytical Chemists [44]. Dry matter was analyzed by drying the samples in an oven at 105 °C to a constant weight. The crude protein was assayed by the Kjeldahl method (N × 6.25) using KjeltecTM 8400 Auto Sample Systems (Foss Tecator AB, Hoganas, Sweden). Crude lipid was measured via the Soxtec extraction method using a solvent extraction system (Soxtec Avanti 2050 Auto System, Foss Tecator AB, Hoganas, Sweden). Ash was measured in the residues of samples burned in a muffle furnace at 550 °C for 8 h.

### 2.5. Determination of Intestinal Permeability and Liver Antioxidant Capacity

Total antioxidant capacity (T-AOC) and malondialdehyde (MDA) content, as well as the activities of glutathion peroxidase (GSH-Px), catalase (CAT) and superoxide dismutase (SOD) in the liver were measured, and the diamine oxidase activity and the concentrations of endotoxin and D-lactic acid in the plasma samples were determined by using commercial assay kits (Nanjing jiancheng Bioengineering Institute, Nanjing, Jiangsu, China), according to the instructions of the manufacturer.

### 2.6. Histological Analysis

After formalin fixation, all of the segments (PI, MI and DI) were removed and cleaned with physiological saline, fixed in Bouin’s solution for 24 h, rinsed with 70% ethanol solution, and finally immersed in 70% ethanol until histological processing was to be performed [45]. The intestinal segments were embedded in paraffin, followed by the cutting of a 5-μm thick slice by using a rotary microtome (KD-2258S, China). The slices were then placed on glass slides and stained with conventional H-E staining for morphometric observations. Pictures were observed under a light microscope (Leica DM5500B, Wetzlar, Germany), and digital images were collected and processed using a digital camera (Leica DFC450) linked to an image-processing software program (Version 4.3.0 Leica). Five slides were used for each segment. A total of 30 measurements were made for each slide in the determinations of the height of mucosal folds (HMF), thickness of muscular layer (TML) and number of mucosal folds (NFM).

### 2.7. TUNEL Assay

The TUNEL staining was performed using a TUNEL Apoptosis Detection Kit (Vazyme Biotech, Nanjing, China) to detect apoptotic cells, according to the manufacturer’s instructions. In brief, histological sections were placed on slides, which was followed by xylene dewaxing treatment, and they were then cleaned twice with distilled water. The rehydrated sections were digested with proteinase K for 4 min at 37 °C, covered with TdT Reaction Buffer for 10 min at 37 °C and then incubated in TUNEL reaction mix containing EdUTP and TdT enzymes for 1 h at 37 °C. After incubation, the sections were washed twice with PBS for 5 min, followed by incubation in streptavidin-HRP and DAB solution for 30 min at 37 °C in the dark. The sections were rinsed with 3% BSA in PBS for 5 min, followed by DAPI counterstain. The slides were viewed under a microscope with a green fluorescence of 520 nm. The areas stained red indicated apoptotic cells.

### 2.8. Intestinal Microbial Diversity

The total DNA was extracted from whole intestinal samples using a DNA extraction kit (Omega Bio-teK, Norcross, GA, USA) according to the manufacturer’s instructions. The integrity, purity and quantity of the DNA samples were detected by 1% (*w*/*v*) agarose gel electrophoresis (Nano Drop 2000, Wilmington, DE, USA), respectively. The V3–V4 region of the 16S rDNA gene of the bacteria was amplified by PCR using the forward primer 338F (5′-ACTCCTACGGGAGCAGCAG-3′) and reverse primer 806R (5′-GGACTACNNGGGTATCTAAT-3′). The operation process of the PCR reaction system was based on our previous research [46]. The original sequence was uploaded to the SRA database of the NCBI. The high-throughput sequencing was carried out using an Illumina Miseq PE300 (Beijing Allwegene technology Co., Ltd., Beijing, China). After constructing a library of small fragments for sequencing, the data were passed through QIIME (v1.8.0) to remove low-quality sequences and chimeras. By comparison with the sliva database, the information of species classification for each OTU was acquired. Alpha diversity analysis, including Shannon, ACE, and Chao1, was performed using Mothur software (version 1.31.2). Based upon the weighted UniFrac distance, cluster analysis was carried out by using a heat map of the R (v3.1.1) software package. The differences of the species communities among samples were compared using the UniFrac algorithm based upon the information of systematic evolution, and by use of beta diversity analysis.

### 2.9. Determination of Gene Expression

Total RNA was isolated from intestinal segment samples using Trizol reagent (Takara Co., Ltd., Nojihigashi, Japan) according to the manufacturer’s instructions. RNA concentration was quantified using a NanoDrop 2000 spectrophotometer (Thermo Scientific, Wilmington, DE, USA). The DNA integrity was detected using 0.1% agarose gel electrophoresis. The RNA samples with OD260/280 ratios greater than 1.9 were chosen for further analysis. In order to remove the vestigial genomic DNA, the RNA was treated with gDNA Eraser. The cDNA from 1 mg total RNA was synthesized using a PrimeScript™ RT reagent Kit (TaKaRa, RR047A, Dalian, China). The cDNA was stored at −80 °C until use. The genes, including inflammatory cytokines, were determined in this study.

The expressions of candidate genes were detected using real-time PCR under an ABI 7500 real-time PCR Detection system (Applied Biosystems, Foster City, CA, USA) using TB GreenTM Rremix Ex TaqTM Ⅱ (Tli RNaseH Plus) (TaKaRa, RR820A, China). The internal reference gene (β-actin) was used to normalize cDNA loading. The expressions of the candidate genes for each sample were assayed in triplicate for accuracy and error estimation based on the previous study [46]. The primer sequences are given in Table 2. All of the primers were provided by an outside company (Shanghai Ruijie Biological Engineering Co., Ltd., Shanghai, China). Confirmation of reaction specificity was performed by melting curve analysis. Standard curves of each of the pairs of primers were set up by plotting Ct values against the log10 of five different dilutions of a cDNA mix solution, in terms of those analyzed samples. Real-time PCR efficiency (E) was calculated by a standard curve originated from the equation E = 1 − 10^(−1/slope) [47]. The expression levels of the originated genes were calculated by the 2^−ΔΔCt^ method [48]. The data for all diets were compared with those of control diet, after verification that the amplified primers were up to an efficiency of approximately 100%.

### 2.10. Statistical Analysis

All data are presented as the mean and standard deviation (SD). The data were analyzed using one-way analysis of variance (ANOVA) to test differences among treatments, followed by the S–N–K multiple comparison test. The normality and homogeneity of variance were confirmed using the Kolmogorov–Smirnov test and Levene’s test in SPSS Statistics 20.0 (SPSS, Michigan Avenue, Chicago, IL, USA), and the data conversion was performed when data were expressed as percentages or ratios prior to statistical analysis. *p* < 0.05 was considered to be statistically significant.

## 3. Results

### 3.1. Growth Performance and Whole-Body Proximate Composition

As shown in Table 3, there was no significant difference in IBW among the four diets. The SBM diet showed the lowest WG and SGR among all the treatments, and the values of the SBM diet were lower (*p* < 0.05) than those of the other diets. The G-4.5 diet exhibited the highest FBW, WG and SGR, and the values were superior (*p* < 0.05) to those of G-10 diet, but were not different from the FM diet. The FR of the SBM diet was inferior (*p* < 0.05) to those of other diets, and no difference was observed in FR among other diets (*p* > 0.05). The HSI of the SBM, G-4.5 and G-10 diets were inferior (*p* < 0.05) to that of the FM diet, and no (*p* > 0.05) difference in HSI was observed between the SBM, G-4.5 and G-10 diets. There were no differences (*p* > 0.05) in FE and survival among treatments.

The change of the whole-body moisture content was opposite to that of the WG, but whole-body lipid content was parallel to that of WG, in response to dietary treatments. The whole-body ash content of G-10 diet was lowest among all treatments, and the value for G-10 was inferior (*p* < 0.05) to that of the FM diet. The whole-body protein content was not (*p* > 0.05) affected by dietary treatments.

### 3.2. Liver Antioxidant Capacity

The BM diet had the lowest liver T-AOC value and GSH-Px activity across treatments (Table 4), and the values were lower (*p* < 0.05) than those of the FM and G-4.5 diets, but were not (*p* > 0.05) different from those of the G-10 diet. The change of liver MDA content was opposite to those of the T-AOC value and GSH-Px activity, in terms of dietary treatments. Liver CAT and SOD activities were not (*p* > 0.05) affected by dietary treatments.

### 3.3. Intestinal Histological Observation

As shown in Figure 1, for the FM diet, the intestinal structure of each layer is clear and complete; the tightly connected single-layer columnar epithelial cells and a large number of intestinal mucosal goblet cells were observed. In comparison with the FM diet, we observed the destroyed intestinal structure of MI and DI of the SBM diet, the irregular arrangement of single-layer columnar epithelial cells, and the reduced number of intestinal mucosal goblet cells, as well as intestinal epithelial cells detached from lamina propria. Compared with the FM diet, both G-4.5 and G-10 diets had a slight damage of intestinal structure. However, the degree of intestinal structure damage of the G-10 diet was greater than that of the G-4.5 diet.

As shown in Table 5, the TML of MI and DI of the FM diet and G-4.5 diet were superior (*p* < 0.05) to that of the SBM diet, but there was no (*p* > 0.05) significant difference between the diets treated with SAP, and the HMF of DI was higher (*p* < 0.05) than those of the G-10 and SBM diets. The TML of DI and the HMF of DI in SBM diet were the lowest among the treatments. However, there was no (*p* > 0.05) difference in the NMF and the HMF in three segments of intestinal mucosa among the diets.

### 3.4. Gene Expression Related to Intestinal Tight Junction Function

The intestinal mRNA levels of *occludin*, *claudin-3* and *ZO-1* genes in the SBM and G-10 diets were down-regulated (*p* < 0.05) compared with the FM diet. However, the mRNA levels for the three genes in the G-4.5 diet were similar (*p* > 0.05) to those in the FM diet (Figure 2).

### 3.5. Intestinal Mucosal Permeability

Figure 3 showed that diamine oxidase activity, D-lactic acid and endotoxin contents in the intestine of G-10 and SBM diets were superior (*p* < 0.05) to those in the FM diet and G-4.5 diet. There were similar (*p* > 0.05) values for the three indices between the G-4.5 diet and the FM diet and between the SBM diet and the G-10 diet.

### 3.6. Intestinal Cell Apoptosis

The results of TUNEL staining showed that the number of intestinal cell apoptosis of the SBM and G-10 diets of DI showed a clear positive signal vs. those of the FM and G-4.5 diets (Figure 4). The SBM diet showed the maximum number of intestinal cell apoptosis among the four diets, followed by the G-10 diet, and the values of the SBM and G-10 diets were greater than those of the FM and G-4.5 diets.

The results of gene expression related to intestinal cell apoptosis are shown in Figure 5. The intestinal expression levels of *caspase-3*, *caspase-8* and *caspase-9* genes were highest in the SBM among the four diets, followed by the G-10 diet, and the values for the three genes of SBM and G-10 diets were superior (*p* < 0.05) to those of the FM diet. However, no differences (*p* > 0.05) in intestinal expression levels of *caspase-3*, *caspase-8* and *caspase-9* genes were observed between FM and G-4.5 diets; the changes of intestinal mRNA levels of *bcl-2* and *bcl-xL* genes were opposite to those of *caspase-3*, *caspase-8* and *caspase-9* genes in response to dietary treatments.

### 3.7. Expression of Intestinal Inflammatory Factor Genes

Figure 6 shows that mRNA levels of the *NF-κB1*, *RelA*, *TAK1*, *IKK*, *MyD88*, *TNF-α*, *IL-1β*, and *IL-8* genes of the SBM and G-10 diets were up-regulated (*p* < 0.05) compared with the FM diet, and the values for the eight genes in the G-4.5 diet were similar (*p* > 0.05) to those of the FM diet. Opposite to the changes of mRNA levels of the eight above-mentioned genes, both the SBM and G-10 diets had down-regulated (*p* < 0.05) expression levels of *IκB*α, *TGF-β1* and *IL-10* genes in comparison with those of the FM diet. The G-4.5 diet had a mRNA level of the three genes similar to that of the FM diet.

### 3.8. Intestinal Microbial Profile

The OTUs, Chao1 and ACE, as well as the Shannon and Simpson indices of the intestine, did not (*p* > 0.05) differ across dietary treatments (Table 6).

As shown in Figure 7A, Proteobacteria (73.2%), Tenericutes (7.0%), Actinobacteria (10.9%), Firmicutes (3.7%), Cyanobacteria (0.9%) and Bacteroidetes (2.2%) were the dominant phyla in the intestines of orange-spotted groupers. Figure 7B shows the relative abundance of the first four dominant phyla among treatments. The relative abundance of Proteobacteria of the SBM diet was superior (*p* < 0.05) to that of the FM diet, but was similar (*p* > 0.05) to those of the G-4.5 and G-10 diets. There were no (*p* > 0.05) significant differences in Actinobacteria and Firmicutes among the four diets.

As shown in Figure 8A, at the genus level, the dominant bacterial genera of the intestine in the four diets were *Achromobacter* (32.6%), *Arthrobacter* (10.5%), *Vibrio* (5.6%), *Bacillus* (2.1%), *Hygromicrobium* (1.3%), *Sediminibacterium* (0.9%), *Chrococcidiopsis* (0.3%) and *Enterovibrio* (0.2%). Figure 8B lists the relative abundance of the first four dominant genera. The relative abundance of *Achromobacter* in the G-10 diet was superior (*p* < 0.05) to that of other diets. The abundance of *Arthrobacter* and *Bacillus* did not (*p* > 0.05) differ across treatments.

## 4. Discussion

Dietary high-SBM inclusion resulted in poor growth performance of fish in the present study and in previous studies [4,12,39,49,50]. In this study, two inclusion levels (4.5% and 10%) of glycinin identical to the SBM substitution for 30% and 60% FM protein, respectively, were used to investigate whether dietary glycinin addition affects the growth performance of orange-spotted grouper. Unexpectedly, 10% glycinin addition achieved better growth rate and feed utilization than did the SBM diet and had a growth rate and feed utilization similar to that of the control, although its growth rate and feed utilization were inferior to that of 4.5% glycinin addition. Similarly, the reduced growth rate and feed utilization were observed in grass carp (*Ctenopharyngodon idella*) [51], hybrid yellow catfish (*Pelteobagrus fulvidraco*♀ × *Pelteobaggrus vachelli*♂) [34] and hybrid grouper [25] when they were fed the diets containing 6–8% glycinin, and the opposite results occurred for those fed the diets containing 2–4% glycinin, as compared with FM diets. These findings indicate that dietary high-glycinin content addition does not constrain growth, and the intermediate-glycinin content addition exhibits superior growth performance to that of the control diet. Our current study also showed that both the 4.5% and the 10% glycinin diets achieved feed intake comparable to the control. However, the hepatosomatic index of fish fed the 4.5% and 10% glycinin diets was lower than that of the control. Furthermore, the lower nutritional status was found to be generally parallel to the higher content of whole-body moisture and lower whole-body contents of crude lipid and ash of fish receiving the SBM diet and that of the 10% glycinin diet when comparing the control. Reduced whole-body ash and lipid contents in fish fed high-SBM diets were also observed as compared with those of FM diets [4,6,12]. Similarly, increased whole-body moisture content [52,53] and lowered whole-body lipid and/or ash contents [52] of fish fed diets containing diets with high glycinin were observed as compared to those of the control. These results indicate that fish fed high-glycinin diets may be suffering from a certain degree of malnutrition, similar to those fish receiving high-SBM diets.

SBM diets showed lower liver values for T-AOC, SOD and GSH-Px, and higher MDA contents in comparison with the FM diet in this study. These reflected high-level hepatic lipid peroxidation caused by high SBM, and indirectly reflected a certain degree of oxidative damage in the liver. The enhanced effect of liver lipid peroxidation in terms of T-AOC value and GSH-Px activity also occurred in fish fed 10% glycinin diets, but not in fish fed 4.5% glycinin diets, which indicates the reduced stress effects of 4.5% glycinin diets compared to 10% glycinin diets. Similarly, 8–12% glycinin diets reduced the T-AOC value and the activities of SOD and/or CAT, and increased MDA contents in the intestine of grass carp [30], Jian Carp (*Cyprinus carpiovar Jian*) [21] and golden crucian carp (*Cyprinus carpio* × *Carassius auratus*) [52]. Therefore, dietary high-glycinin contents could enhance the hepatic and intestinal lipid peroxidation of fish.

It is well known that the intestine is the place for fish to digest and absorb nutrients [7]. The intestinal histomorphological integrity is the prerequisite for keeping a good digestion and absorption state, and intestinal disease resistance is a crucial barrier against exogenous pathogens in the immune defense of fish [46]. The height and length of intestinal plica and the TML are important indicators for measuring the intestinal digestion and absorption function. In previous studies, 8% glycinin diets reduced the plica heights of PI and MI of Jian carp [21], and the TML and HMF of DI of *Rhynchocypris lagowskii* [53] vs. FM diets, which was supported by our current results and previous results that 10% glycinin diets reduced the TML and HMF of DI of hybrid grouper [40] and hybrid yellow catfish [34]. The findings showed the MI and DI were more susceptible to glycinin than was the PI. However, the HMF and TML were greater in fish fed 2% glycinin diets than in fish fed 8% glycinin diets [25]. The above observations showed that dietary high-glycinin content inclusion could potentially cause adverse alterations in the intestinal morphology of fish, thereby affecting the absorption of nutrients by the intestine.

Studies have shown that dietary high-SBM can also damage the intestinal structural integrity of fish [12,54], leading to the occurrence of SBM-induced enteritis, and that the severity of enteritis correlates in a dose-dependent manner with increasing SBM levels. In this study, the effects of dietary glycinin on the intestinal structural integrity were determined through the tight junction function in the intestine of grouper at the molecular level. The down-regulated expression levels of *claudin-3*, *occludin* and/or *ZO-1* genes were observed in the SBM and 10% glycinin diets in our current study, and in the 8% glycinin diets [21,26] and high-SBM diets [13,14] in previous studies in comparison with the FM diets, indicating that dietary high-glycinin contents could damage the intestinal intercellular structure by disrupting the intestinal tight junction structure of fish, including grouper [26].

As an important barrier against external bacterial invasion, the intestine plays an extremely significant role in maintaining animal homeostasis [55]. In this study, both SBM diets and 10% glycinin diets increased plasma diamine oxidase activity, D-lactic acid and endotoxin contents. The enhancement effect of high-glycinin (8% glycinin) content addition on the indices was also observed in previous studies with grass carp [26] and *Rhynchocypris lagowskii* [56,57]. This indicates that the damage of the intestinal mucosal barrier is associated with the increased permeability of intestinal mucosa of fish caused by high-glycinin contents. This may be a secondary effect of dietary high glycinin, causing foodborne allergic reactions in fish. However, the intermediate glycinin level (4.5% glycinin) did not trigger the adverse effect in orange-spotted grouper.

The Caspase family mediates intestinal cell apoptosis induced by dietary high-glycinin content inclusion [57]. The dietary high-glycinin (6–8%) contents were found to trigger mRNA expression of *caspase-3*, *caspase-8* and/or *caspase-9* genes in MI for grass carp [26], and in DI for Jian carp [21] and hybrid yellow catfish [34]. In the present study, we determined the effects of dietary glycinin on intestinal apoptosis of orange-spotted groupers by using TUNEL fluorescence staining and gene expression levels of intestinal *caspase-3*, *caspase-8* and *caspase-9*. We observed that fish fed SBM and 10% glycinin diets exhibited an increase in the number of intestinal apoptotic cells and an up-regulation of intestinal expression levels of *caspase-3*, *caspase-8* and *caspase-9* genes vs. those fish fed the FM diets, but did not observe this effect on fish fed 4.5% glycinin diets. Therefore, apoptosis of intestinal cells of fish could be induced by dietary high-glycinin contents by promoting the intestinal caspases activation but not by intermediate-glycinin contents.

In addition to the caspase-mediated intestinal apoptosis, we also conducted further research on the NF-κB signaling pathway regulating intestinal apoptosis. The activated NF-κB can up-regulate the expression levels of *bcl-2* and *bcl-xL* genes, thereby leading to the reduction of mitochondrial membrane permeability and obstruction of cytochrome C release, and thus inhibits apoptosis [58]. The cytokine TNF-α acts as an activator of the NF-κB signaling pathway. The up-regulation of TNF-α will stimulate *NF-κB1*/*RelA* to enter the nucleus and activate those genes containing DNA binding sites for NF-κB. This will lead to apoptosis occurrence, which in turn induces down-regulation of expressions of downstream *bcl-2* and *bcl-xL* genes [59]. In this study, the mRNA expressions of *NF-κB1* and *TNF-α* genes were up-regulated, but the mRNA expressions of *bcl-2* and *bcl-xL* genes were down-regulated by SBM and 10% glycinin diets vs. the FM diets. Similarly, down-regulated intestinal *bcl-2* mRNA expressions were observed in grass carp [26] and hybrid yellow catfish [34] that received 8% glycinin diets compared with those fish fed the control diets. The results indicate that the occurrence of intestinal apoptosis triggered by dietary high-glycinin content addition is associated with the down-regulated expression of anti-apoptosis genes *bcl-2* and *bcl-xL* and up-regulated expression of pro-apoptosis genes *TNF-α* and/or *caspase-3*, both in high-SBM and high-glycinin diets [8,57], by activation of the NF-κB signaling pathway.

There is an interdependent relationship between intestinal microbiota and the host, and intestinal microbiota is inseparable from nutrition, immunity and metabolism, forming a relatively independent intestinal microbiota homeostasis [60]. The abundance levels of the intestinal microbial communities of fish are influenced by various environmental factors such as the source, composition, and type of feed, etc. [61]. For example, at the phylum level, the decreased Firmicutes and Actinobacteria abundance and the increased Cyanobacteria and Tenericutes abundance, as well as the lowered Fictibacillus Lactobacillus abundance and increased Mycoplasma abundance at the genus level were observed in the intestine of largemouth bass (*Micropterus salmoides*) fed high-SBM diets vs. the control diets [9]. In this study, the intestinal dominant phyla of fish were Proteobacteria (73.2%), Tenericutes (7.0%), Actinobacteria (10.9%), Firmicutes (3.7%), Bacteroidetes (2.2%) and Cyanobacteria (0.9%). Interestingly, like SBM diets, 10% glycinin diets generally promoted the abundance of phylum Proteobacteria in the intestine of orange-spotted grouper vs. FM diets. The increased intestinal Proteobacteria abundance was also observed in largemouth bass fed SBM diets [62] and in hybrid grouper fed 10% glycinin diets [40]. Proteobacteria has many pathogenic bacteria that can secrete a large amount of lipopolysaccharides [63], and higher lipopolysaccharides levels exacerbate stroke by activating NF-κB and thus produce large amounts of pro-inflammatory cytokines [64]. On the contrary, the intestinal Tenericutes abundance was superior in FM and 4.5% glycinin diets to those of the SBM and 10% glycinin diets. It is still unknown whether Tenericutes belongs to a pathogenic bacterial at the phylum level in fish, a question which requires further research [65]. The intestinal phylum Proteobacteria enrichment and phylum Tenericutes reduction may indicate the imbalance of intestinal microbiota [66]. However, differing from our current observations, 10% glycinin diets tended to reduce the abundance of Bacteroides, Actinobacteria and Firmicutes, though not differently from the FM diet [40]. The variation in intestinal bacterial abundance between spotted-orange and hybrid grouper may be due to differences in fish species and feed composition.

In the present study, SBM and 10% glycinin diets generally promoted the abundance of genus *Achromobacter* and/or *Vibrio* in the intestine of orange-spotted grouper, in comparison with FM diets. The increased abundance of genus *Vibrio* in the intestine of hybrid grouper was observed [40]. The genus *Achromobacter* is considered opportunistic pathogens of low virulence in humans [67] and the genus *Vibrio*, such as *Vibrio anguallanim*, is another opportunistic pathogen that can cause hemorrhagic septicemia and vibriosis in many fish species [68]. Furthermore, the intestinal abundance was highest, respectively, in SBM diets for the former and in 10% glycinin diets for the latter among the treatments (Figure 8A) in this study. This might partly explain the reason for the induction of enteritis of fish caused by dietary high-SBM contents and glycinin addition through providing favorable conditions for opportunistic pathogens to colonize the intestine. In addition, the unidentified bacterial genera in SBM diets were higher than those in other groups, which may be another trigger for enteritis in the case of feeding high-SBM contents.

It is clear that that the intestine of fish is also an important organ that performs immune defense functions. Under the stimulation of pathogens, a series of complex intestinal inflammatory reactions are triggered and activated through cytokines and complementary factors secreted by macrophages and leukocytes [69]. NF-kB as a central mediator of pro-inflammatory gene induction also participates in the inflammation regulation through inducing the expression of various pro-inflammatory genes, including those encoding *TNF-α*, *IL-1β*, *IL-6*, *IL-12* and cyclooxygenase-2 [70]. TAK1 activates the downstream IKK complex and phosphorylates the NF-κB inhibitor IκBα, thereby mediating NF-κB activation [71]. In addition, NF-κB activation mediates immune and inflammatory responses triggered by MyD88 [72]. The NF-κB dimer RelA undergoes rapid and transient nuclear translocation under the action of phosphorylated IκBα via IKK activation [71]. Inflammatory responses are associated with aberrant T cell activation due to aberrant activation of NF-κB [73]. After activation, CD4+T cells develop into effector T cells (inflammatory T cells), which secrete cytokines and mediate aberrant inflammatory responses [74].

Previous studies showed that dietary high-SBM content triggers significant changes in the expression of inflammatory-factor genes in fish [7,12,46]. Massive production of pro-inflammatory cytokines was observed in fish fed high-SBM diets [8,11,13]. In contrast, the down-regulated expressions of anti-inflammatory genes *IL-10* and *TGF-β* were observed in the liver and spleen of fish fed high-SBM diets [14]. In the present study, the SBM and 10% glycinin diets up-regulated the expression levels of *TNF-α*, *IL-1β*, *IL-8*, *MyD88*, *IKK-α*, *TAK1*, *RelA* and *NF-κB1* genes, but down-regulated the expression levels of *IκBα*, *TGF-β1* and *IL-10* genes, compared to FM diets. Similarly, an up-regulation of pro-inflammatory genes, including *TNF-α*, *IL-1β* and/or *IL-8*, and a down-regulation of anti-inflammatory genes *IL-10* and/or *TGF-β* in the intestine were observed in recent studies with fish fed 8% glycinin diets [21,26,53,56]. The above results show that NF-κB induces the expression of pro-inflammatory genes and also regulates inflammatory responses caused by high glycinin diets.

## 5. Conclusions

In conclusion, dietary high-glycinin (10%) content addition, but not that of intermediate-glycinin (4.5%) content, could reduce the growth performance, accompanied by the enhancement of the hepatic lipid peroxidation. In the meantime, 10% glycinin in the feed could increase mucosal permeability and damage structural integrity in the intestine by disrupting the tight junction structure and inducing apoptosis in the intestine through promotion of the activation of intestinal caspases. With these changes, intestinal dysbacteriosis and inflammatory reactions occur, a situation which further develops into enteritis, thus reducing the digestion and absorption of nutrients.

## Figures and Tables

**Figure 1 animals-13-02605-f001:**
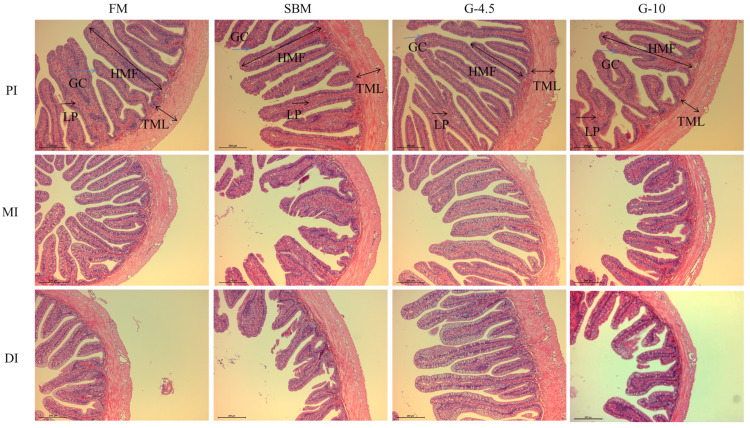
Histological observation of proximal intestine (PI), mid intestine (MI), and distal intestine (DI) of fish fed the experimental diets across a 56-day growth trial period (HE × 100). PI, MI and DI represent the fore, mid and hind guts, respectively; the black two-way arrows refer to the thickness of muscular layer (TML) and the height of mucosal folds (HMF); the black one-way arrows refer to the lamina propria (LP); the blue single arrows refer to goblet cells (GC). FM, fish meal diet (FM diet); SBM, 30% FM was replaced by soybean meal in FM diets, without glycinin supplementation; G-4.5 and G-10, 4.5% and 10% glycinin were added in FM diets, respectively.

**Figure 2 animals-13-02605-f002:**
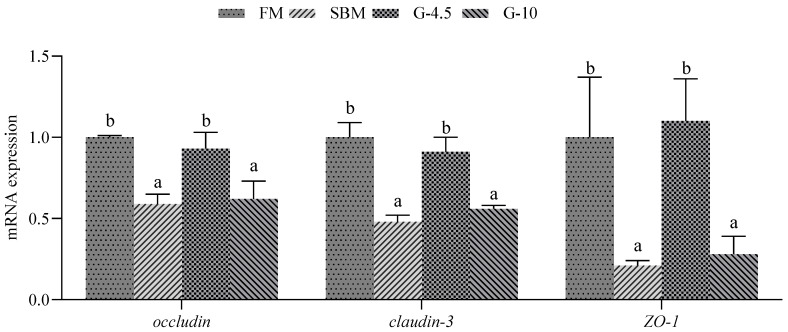
The expression of genes related to intestinal tight junction functions in the fish fed the experimental diets across a 56-day growth trial period. Data are shown as means ± SD (n = 3 tanks) and analyzed by one-way ANOVA. Bars for each gene having various superscripts show significant differences (*p* < 0.05), while those having the same letter or no letter superscripts show no significant differences (*p* > 0.05). FM, fish meal diet (FM diet); SBM, 30% FM was replaced by soybean meal in FM diets, without glycinin supplementation; G-4.5 and G-10, 4.5% and 10% glycinin were added in FM diets, respectively. ZO-1, zonula occludens-1.

**Figure 3 animals-13-02605-f003:**
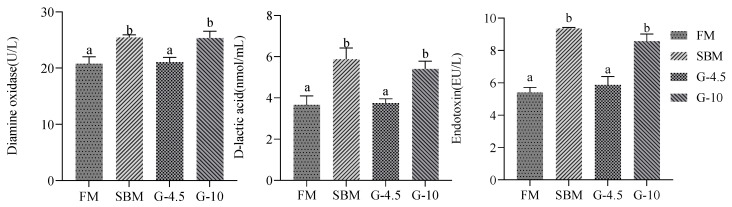
Blood components in regard to intestinal mucosal permeability in the fish fed the experimental diets across a 56-day growth trial period. Data are shown as means ± SD (n = 3 tanks) and analyzed by one-way ANOVA. Bars for each indicator having various superscripts show significant differences (*p* < 0.05), while those having the same letter or no letter superscripts show no significant differences (*p* > 0.05). FM, fish meal diet (FM diet); SBM, 30% FM was replaced by soybean meal in FM diets, without glycinin supplementation; G-4.5 and G-10, 4.5% and 10% glycinin were added in FM diets, respectively.

**Figure 4 animals-13-02605-f004:**
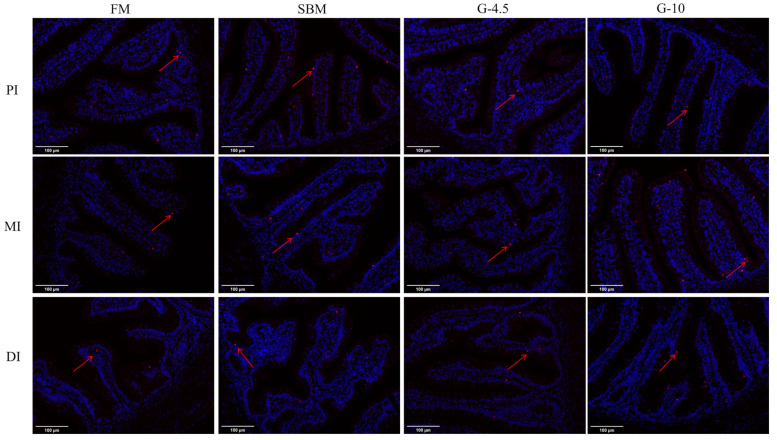
Changes of middle-intestinal-cell apoptosis in fish fed the experimental diets across a 56-day growth trial period. The red arrow indicate that location of apoptotic cells. PI, MI and DI represent the proximal, middle and distal intestines, respectively; red particles refer to apoptotic cells. FM, fish meal diet (FM diet); SBM, 30% FM was replaced by soybean meal in FM diets, without glycinin supplementation; G-4.5 and G-10, 4.5% and 10% glycinin were added in FM diets, respectively.

**Figure 5 animals-13-02605-f005:**
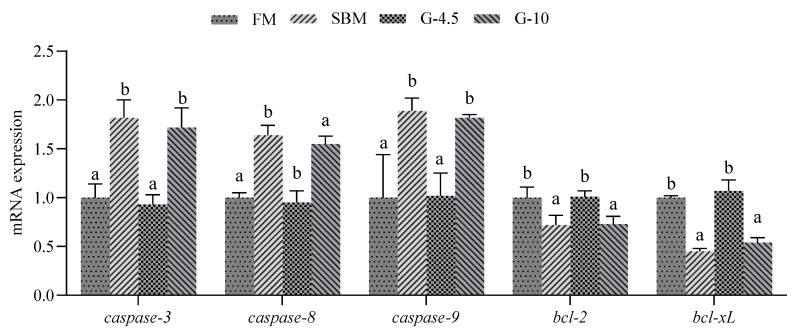
The expression of intestinal cell apoptosis genes in fish fed the experimental diets across a 56-day growth trial period. Data are shown as means ± SD (n = 3 tanks) and analyzed by one-way ANOVA. Bars for each gene having various superscripts show significant differences (*p* < 0.05), while those having the same letter or no letter superscripts show no significant differences (*p* > 0.05). FM, fish meal diet (FM diet); SBM, 30% FM was replaced by soybean meal in FM diets, without glycinin supplementation; G-4.5 and G-10, 4.5% and 10% glycinin were added in FM diets, respectively. *caspase*, cyteine aspartic acid specific protease; *bcl*, B cell lymphoma.

**Figure 6 animals-13-02605-f006:**
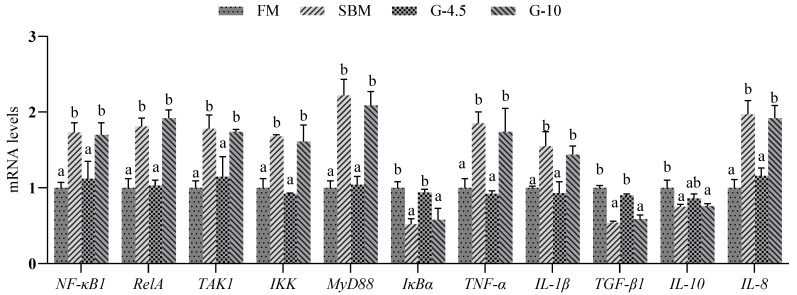
The expression of intestinal inflammatory factor genes in the fish fed the experimental diets across a 56-day growth trial period. Data are shown as means ± SD (n = 3 tanks) and analyzed by one-way ANOVA. Bars for each gene having various superscripts show significant differences (*p* < 0.05), while those having the same letter or no letter superscripts show no significant differences (*p* > 0.05). FM, fish meal diet (FM diet); SBM, 30% FM was replaced by soybean meal in FM diets, without glycinin supplementation; G-4.5 and G-10, 4.5% and 10% glycinin were added in FM diets, respectively. *NF-κB1* = nuclear factor kappa-B1; *RelA* = transcription factor p65; *TAK1* = transforming growth factor activated kinase-1; *IKK* = inhibitor of kappa B kinase; *MyD88* = myeloid differentiation factor 88; *IκBα* = inhibitor of κB alpha; *TNF-α* = tumor necrosis factor-alpha; *IL-1β* = interleukin-1 beta; *TGF-β1* = transforming growth factor-beta 1; *IL =* interleukin.

**Figure 7 animals-13-02605-f007:**
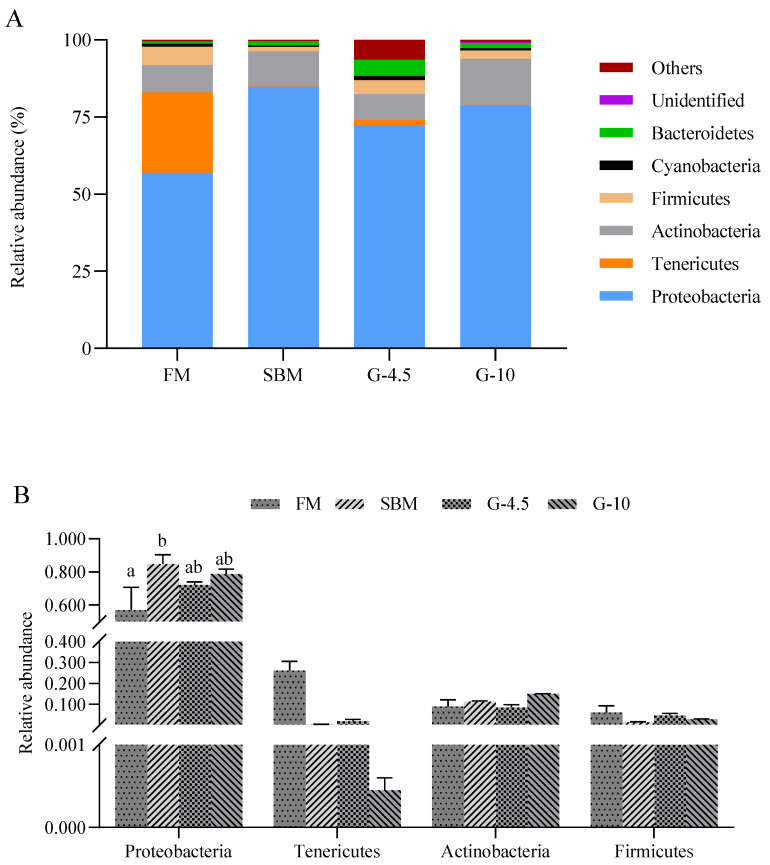
The intestine microbiota at phylum level in the fish fed the experimental diets across a 56-day growth trial period. (**A**) Relative abundance of the distal intestine microbiota and (**B**) relative abundance for the top four phylum levels. Data are shown as means ± SD (n = 3 tanks) and analyzed by one-way ANOVA. Bars for each bacterium having various superscripts show significant differences (*p* < 0.05), while those having the same letter or no letter superscripts show no significant differences (*p* > 0.05). FM, fish meal diet (FM diet); SBM, 30% FM was replaced by soybean meal in FM diets, without glycinin supplementation; G-4.5 and G-10, 4.5% and 10% glycinin were added in FM diets, respectively.

**Figure 8 animals-13-02605-f008:**
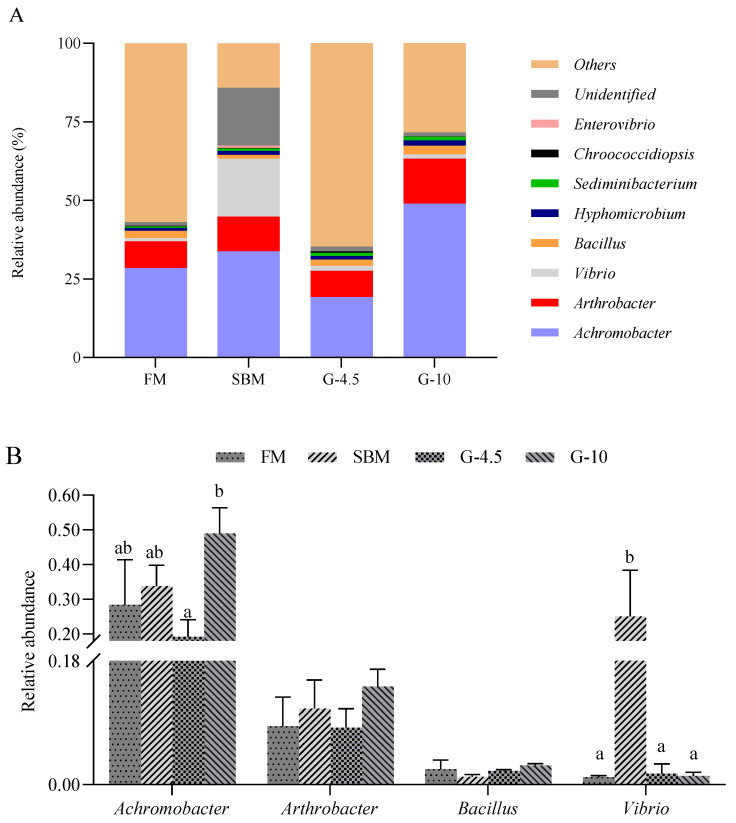
The relative abundance of intestine microbiota at genus levels in the fish fed the experimental diets across a 56-day growth trial period. (**A**) Relative abundance of the intestine microbiota and (**B**) relative abundance for the top four genus levels. Data are shown as means ± SD (n = 3 tanks) and analyzed by one-way ANOVA. Bars for each bacterium having various superscripts show significant differences (*p* < 0.05), while those having the same letter or no letter superscripts show no significant differences (*p* > 0.05). FM, fish meal diet (FM diet); SBM, 30% FM was replaced by soybean meal diet in FM diets, without glycinin supplementation; G-4.5 and G-10, 4.5% and 10% glycinin were added in FM diets, respectively.

**Table 1 animals-13-02605-t001:** Ingredients and proximate composition of experimental diets (on an as-fed basis, g/kg).

Items	Diets ^1^			
	FM	SBM	G-4.5	G-10
Ingredients				
Fish meal ^2^	520	220	520	520
Casein	109.2	105.1	72.3	27.2
Gelatin	27.3	26.3	18.1	6.8
Soybean meal	0	470	0	0
Fish oil	8.2	35.2	8.2	8.2
Soybean oil	35	35	35	35
Soybean lecithin	20	20	20	20
Glycinin	0	0	45	100
Corn starch	245.1	54.2	247.2	248.6
Vitamin mix ^3^	4	4	4	4
Mineral mix ^4^	5	5	5	5
Stay-C 35%	0.2	0.2	0.2	0.2
Sodium alginate	10	10	10	10
Ca(H_2_PO_4_)_2_	15	15	15	15
Nutrient level of diets (analyzed values)				
Dry matter	917.3	912.5	919.1	921.2
Crude protein	479.3	481.2	482.7	479..1
Crude lipid	117.7	115.9	113.4	114.7
Ash	155	101.4	138.6	138.9
Gross energy (kJ/g)	19.27	20.06	19.53	19.61

^1^ FM, fish meal diet (FM diet); SBM, 30% FM was replaced by soybean meal in FM diets, without glycinin supplementation; G-4.5 and G-10, 4.5% and 10% glycinin were added in FM diets, respectively. ^2^ Fish meal was obtained from Trident Seafoods Corporation, Akutan, AK, USA, crude protein: 69.01%, crude lipid: 9.06%; Soybean meal was obtained from Xiamen Xiangyu Shengzhou Oil Co., Ltd., Xiamen, China crude protein: 45.50%, crude lipid: 0.72%; Glycinin products came from the laboratory of Professor Guo Shuntang of China Agricultural University. The purity of the glycinin product was 79.97%. ^3^ Vitamin premix and ^4^ Mineral premix were based on Qin et al. [41].

**Table 2 animals-13-02605-t002:** Primers for the intestinal genes.

Genes	Forward (5′-3′)	Reverse (5′-3′)	Amplicon Size (bp)	E-Value (%)	Accession No.
*caspase-3*	GATGCTGCTGCTGCTATGC	GCCGTCAGTGCCGTATATTATC	189	99	XM_033633485
*caspase8*	TGCCTTGGTGGTATGCGTGCT	GGTGAAGGGCGAGGTCAGTTCT	100	98	XM_033649666.1
*caspase9*	GCCTGTGGAGGAGGTGAAAGAGA	GCTGCTGGATGACATCGGAATGG	114	103	XM_033629367.1
*bcl-2*	GTGCGTGGAGTGCGTTGAGAA	CGCTCCCATCCTCTTTGGCTCT	120	99	KY321170.1
*bcl-xL*	AGTAACGGCTTGCTGGTCAA	GCTGTGGTAGGCTGTGTCA	192	98	MH513638.1
*occludin*	GGCTACGGTGATCGTGTTGTGT	CCGCCTCCATAACCTCCTCCAT	174	101	XM_033622283.1
*claudin3*	GCATTGACGACGAGGCATCCAA	GCCGACCAGGAGACAGGAATGA	100	106	MK782153.1
*ZO-1*	CGGCAGATCAGCAATGGCAACC	TGGTTCAGGCAGCGGAGGTAAC	165	95	MK809396.1
*NF-κB1* (*P50*)	CTTACATTCGCCGCCTCAGT	TGCAACAACGCCTTCAAACC	158	103	JX856139.1
*RelA* (*P65*)	TCTTCTCAGTCCAGCCCAAGGT	GGTGGTAGAGGAGCAGGAGGAT	167	96	EU219847.1
*MyD88*	GCATTGACGACGAGGCATCCAA	GCCGACCAGGAGACAGGAATGA	100	98	JF271883
*IKK-α*	TGGCTGAGAGCGAACAAGTCCT	AGCAGAGGCGGCACTGAAGAT	151	99	KM669150.1
*TAK1*	TCTCAAGGGAGCAACGACAC	GCAGGCAGACTCTCAACACT	122	104	JX856141
*IL-8*	AAGTTTGCCTTGACCCCGAA	TGAAGCAGATCTCTCCCGGT	101	94	FJ913064.1
*IL-1β*	GCAACTCCACCGACTGATGA	ACCAGGCTGTTATTGACCCG	107	116	EF582837.1
*TNF-a*	GGATCTGGCGCTACTCAGAC	CGCCCAGATAAATGGCGTTG	135	91	FJ009049.1
*IL-10*	GTCCACCAGCATGACTCCTC	AGGGAAACCCTCCACGAATC	124	99	KJ741852.1
*TGF-β1*	GCTTACGTGGGTGCAAACAG	ACCATCTCTAGGTCCAGCGT	112	102	GQ503351.1
*β-actin*	GATCTGGCATCACACCTTCT	CATCTTCTCCCTGTTGGCTT		104	AY510710.2

**Table 3 animals-13-02605-t003:** Effects of dietary glycinin addition on the growth performance and whole-body proximate composition of orange-spotted groupers across a 56-day growth trial period ^1^.

Items	Diets ^2^			
	FM	SBM	G-4.5	G-10
Growth performance				
IBW (g/fish) ^3^	8.32 ± 0.13	8.19 ± 0.26	8.27 ± 0.09	7.94 ± 0.23
FBW (g/fish) ^3^	61.52 ± 2.43 ^c^	43.91 ± 1.11 ^a^	63.96 ± 0.65 ^c^	57.50 ± 0.55 ^b^
WG (%) ^3^	639.3 ± 18.0 ^bc^	436.9 ± 15.3 ^a^	673.9 ± 9.0 ^c^	625.4 ± 14.6 ^b^
SGR (%/d) ^3^	3.57 ± 0.04 ^bc^	3.00 ± 0.05 ^a^	3.66 ± 0.02 ^c^	3.54 ± 0.04 ^b^
FE (%) ^3^	110 ± 3	107 ± 5	115 ± 3	105 ± 2
FR (%/d)	2.24 ± 0.03 ^ab^	2.14 ± 0.04 ^a^	2.21 ± 0.04 ^ab^	2.19 ± 0.03 ^ab^
Survival (%) ^3^	100	100	100	100
HSI (%) ^4^	3.02 ± 0.13 ^b^	2.44 ± 0.04 ^a^	2.57 ± 0.14 ^a^	2.39 ± 0.09 ^a^
Proximate composition (%)				
Moisture	67.61 ± 0.27 ^a^	71.08 ± 0.51 ^c^	68.62 ± 0.80 ^ab^	70.53 ± 0.35 ^c^
Crude protein	18.08 ± 0.47	17.48 ± 0.18	18.35 ± 0.07	17.63 ± 0.13
Crude lipid	6.58 ± 0.20 ^bc^	5.75 ± 0.16 ^a^	6.81 ± 0.10 ^c^	6.24 ± 0.13 ^ab^
Ash	4.66 ± 0.10 ^c^	4.30 ± 0.09 ^ab^	4.43 ± 0.07 ^bc^	4.18 ± 0.07 ^a^

^1^ Statistical analysis was performed by one-way ANOVA, followed by the S–N–K test. ^2^ FM, fish meal diet (FM diet); SBM, 30% FM was replaced by soybean meal in FM diets, without glycinin supplementation; G-4.5 and G-10, 4.5% and 10% glycinin were added in FM diets, respectively. ^3^ Values are shown as the means ± SD (n = 3 tanks). ^4^ Values are shown as the means ± SD (n = 20 fish). Values in a line having various superscripts show significant differences (*p* < 0.05), while those having the same letter or no letter superscripts show no significant differences (*p* > 0.05). IBW, initial body weight (g/fish); FBW, final body weight (g/fish); WG, weight gain (%) = 100 × (FBW − IBW)/IBW; SGR, specific growth rate (%/d) = 100 × (ln FBW − ln IBW)/days; FE, feed efficiency (%) = 100 × (FBW − IBW)/feed intake (as fed basis, g/fish); FR, feeding rate (%/d) =100 × (feed intake /((FBW + IBW) /2 × days)); Survival (%) = 100 × final fish number/initial fish number; HSI, hepatosomatic index (%) = 100 × liver weight (g/fish)/body weight (g/fish).

**Table 4 animals-13-02605-t004:** Effects of dietary glycinin addition on liver antioxidant indices of orange-spotted groupers across a 56-day growth trial period ^1^.

Items	Diets ^2^			
	FM	SBM	G-4.5	G-10
T-AOC (mmol/g)	0.38 ± 0.01 ^b^	0.25 ± 0.06 ^a^	0.37 ± 0.02 ^b^	0.28 ± 0.04 ^ab^
GSH-Px (U/mg prot)	62.02 ± 1.13 ^b^	54.80 ± 0.71 ^a^	62.75 ± 2.61 ^b^	58.82 ± 1.66 ^ab^
CAT (U/mg prot)	7.02 ± 1.53	7.39 ± 1.00	8.37 ± 1.52	7.87 ± 0.89
SOD (U/mg prot)	204.43 ± 3.02	190.61 ± 8.22	198.85 ± 7.08	197.89 ± 9.97
MDA (nmol/mg prot)	3.08 ± 0.43 ^a^	6.27 ± 0.45 ^b^	3.17 ± 0.39 ^a^	4.10 ± 0.32 ^a^

^1^ Data are shown as means ± SD (n = 3 tanks). Statistical analysis was performed by one-way ANOVA, followed by the S–N–K test. ^2^ FM, fish meal diet (FM diet); SBM, 30% FM was replaced by soybean meal in FM diets, without glycinin supplementation; G-4.5 and G-10, 4.5% and 10% glycinin were added in FM diets, respectively. Values in a line having various superscripts show significant differences (*p* < 0.05), while those having the same letter or no letter superscripts show no significant differences (*p* > 0.05). T-AOC, total antioxidant capacity; GSH-Px, glutathion peroxidase; SOD, superoxide dismutase; MDA, malondialdehyde; CAT, catalase.

**Table 5 animals-13-02605-t005:** Histological examinations of the proximal intestine (PI), middle intestine (MI) and distal intestine (DI) of fish fed the experimental diets across a 56-day growth trial period (magnification × 200) ^1^.

Items ^3^		Diets ^2^			
		FM	SBM	G-4.5	G-10
PI	HMF (μm)	449.89 ± 19.40	421.35 ± 22.14	455.80 ± 20.34	428.75 ± 14.84
	TML (μm)	93.50 ± 3.90	71.47 ± 10.56	90.37 ± 15.90	84.44 ± 7.67
	NFM (unit)	40.00 ± 0.58	37.67 ± 1.76	41.33 ± 1.86	41.33 ± 1.86
MI	HMF (μm)	370.86 ± 25.48	323.06 ± 12.57	384.08 ± 34.42	354.67 ± 23.82
	TML (μm)	73.91 ± 2.64 ^b^	61.47 ± 4.50 ^a^	72.84 ± 4.29 ^b^	69.72 ± 4.13 ^ab^
	NFM (unit)	28.67 ± 0.33	29.00 ± 0.58	29.00 ± 1.15	31.33 ± 1.76
DI	HMF (μm)	341.18 ± 9.57 ^b^	279.84 ± 15.43 ^a^	342.93 ± 10.73 ^b^	305.48 ± 9.41 ^a^
	TML (μm)	77.67 ± 1.67 ^b^	64.79 ± 3.59 ^a^	76.41 ± 2.38 ^b^	71.46 ± 3.65 ^ab^
	NFM (unit)	31.33 ± 0.33	32.00 ± 1.00	33.67 ± 1.86	34.00 ± 2.31

^1^ Data are shown as means ± SD (n = 3 tanks) and analyzed by one-way ANOVA. ^2^ Values in a line having various superscripts show significant differences (*p* < 0.05), while those having the same letter or no letter superscripts show no significant differences (*p* > 0.05). ^3^ PI, proximal intestine; MI, middle intestine; DI, distal intestine; HMF, height of mucosal folds; TML, thickness of muscular layer; NFM, number of mucosal folds. FM, fish meal diet (FM diet); SBM, 30% FM was replaced by soybean meal in FM diets, without glycinin supplementation; G-4.5 and G-10, 4.5% and 10% glycinin were added in FM diets, respectively.

**Table 6 animals-13-02605-t006:** Richness and diversity indices of the microbial community in intestinal samples of orange-spotted groupers fed the experimental diets across a 56-day growth trial period ^1^.

Items	Diets ^2^			
	FM	SBM	G-4.5	G-10
OTUs	245.00 ± 80.16	228.33 ± 113.87	166.33 ± 51.94	255.67 ± 113.91
Ace	473.72 ± 12.88	476.39 ± 85.51	514.60 ± 80.80	339.86 ± 29.76
Chao1	457.24 ± 28.78	380.55 ± 131.81	493.16 ± 93.57	253.60 ± 53.60
Shannon	2.81 ± 0.4	3.29 ± 0.16	4.36 ± 0.54	3.45 ± 0.90
Simpson	0.70 ± 0.07	0.80 ± 0.01	0.75 ± 0.14	0.75 ± 0.14

^1^ Data are shown as means ± SD (n = 3 tanks). Statistical analysis was performed by one-way ANOVA, followed by the S–N–K test. ^2^ FM, fish meal diet (FM diet); SBM, 30% FM was replaced by soybean meal in FM diets, without glycinin supplementation; G-4.5 and G-10, 4.5% and 10% glycinin were added in FM diets, respectively. Values in a line having different superscripts show significant differences (*p* < 0.05), while those having the same letter or no letter superscripts show no significant differences (*p* > 0.05). OTUs, operation altaxonomic units; Ace, abundance-based coverage estimator; Chao1, chao1 richness estimator; Shannon, Shannon diversity index; Simpson, Simpson diversity index.

## Data Availability

The data that support the findings of this study are available from the corresponding author upon reasonable request.

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
