# Peer review of "Dietary High Glycinin Reduces Growth Performance and Impairs Liver and Intestinal Health Status of Orange-Spotted Grouper (Epinephelus coioides)"

_animals, 2023, doi:10.3390/ani13162605_

Round 1

Reviewer 1 Report

The study is generally well-designed and can attract readers' attention in this field. However, there are some problems in this study. Some of these are as follows.

-One of the big problems is the inaccurate interpretation of the growth results, and the inaccuracy this reveals in the title of the study. According to Table 3, this statement is not entirely correct. Let me explain when the growth parameter values of the control group (FM) and high-dose glycine (G-10) groups are compared, it is seen that there is a decrease in all parameter values in the G-10 group. In fact, this decrease in FBM and HSI parameters seems to be statistically significant. Therefore, it is not entirely correct to use such an expression, especially in the title. The title needs to be corrected appropriately. In addition, necessary adjustments should be made in the text in this direction.

-Which hypothesis is being tested in the study? The purpose of the study should be clearly stated.

-Oxidative stress is the most important part of this study. However, the importance of oxidative stress for fish in the introduction? The reason? and I think it would be useful to give some sample studies. Thus, the work appeals to a wider audience. Please see attached file.

-Could this decrease in casein and gelatin in diets cause extra stress in fish?(Page 3)

-(Page 5) How much (grams) of feed was fed each time?
Was this amount sufficient to feed the 20 fish in each aquarium?
How did you determine this amount?

-(Page 5) About FI-    There is no data about them (FI) in the text. What are these values? These values should be added to the result section.

-Is there a special reason for choosing the liver for the study?

Other suggestions are in the attached file.

Author Response

Reviewer 1
The study is generally well-designed and can attract readers' attention in this field. However, there are some problems in this study. Some of these are as follows.

  1. One of the big problems is the inaccurate interpretation of the growth results, and the inaccuracy this reveals in the title of the study.According to Table 3, this statement is not entirely correct. Let me explain when the growth parameter values of the control group (FM) and high-dose glycine (G-10) groups are compared, it is seen that there is a decrease in all parameter values in the G-10 group. In fact, this decrease in FBM and HSI parameters seems to be statistically significant. Therefore, it is not entirely correct to use such an expression, especially in the title. The title needs to be corrected appropriately. In addition, necessary adjustments should be made in the text in this direction.

Response: Thank you for your valuable comments. We revised the title of the article and made adjustments in the text in the direction.

  1. line 53-55:It would be more appropriate to support this statement with the reference below.

-S Yedier, E Gümüs, EJ Livengood, FA Chapman 2014. The relationship between carotenoid type and skin color in the ornamental red zebra cichlid Maylandia estherae. AACL Bioflux 7( 3) : 207- 216.2014

Response: As suggested by the reviewer, we replaced the references.

  1. Oxidative stress is the most important part of this study. However, the importance of oxidative stress for fish in the introduction? The reason? and I think it would be useful to give some sample studies. Thus, the work appeals to a wider audience. Please see attached file.

Response: We sincerely appreciate the valuable comments. We have added the description on oxidative stress in fish and relevant references.

  1. Which hypothesis is being tested in the study? The purpose of the study should be clearly stated.

Response: We have re-written this part according to the reviewer’s suggestion.

  1. Could this decrease in casein and gelatin in diets cause extra stress in fish?(Page 3).

Response: Thanks for your comments. I don’t think so. Besides fish meal, casein and gelatin are protein ingredients with relatively high quality protein. They are usually used in the research on the nutrition and feed utilization of fish. In order to ensure the good nutritional value of casein and gelatin when they are used in the feed formulations, the ratio of casein to gelatin in the feeds is between 4:1 and 5:1, which is an appropriate proportion to realize a better AA balance and good utilization of the compound feeds. The casein and gelatin are also usually used as main protein sources to prepare purified or semi-purified diets in the determination for the requirements of protein, amino acids, vitamins of fish, etc. The adopted ratio of casein to gelatin in the experimental diets is 4:1 in the present study.

  1. (Page 5) How much (grams) of feed was fed each time?
    Response: We usually adopt the feeding strategy that is somewhat like terrestrial animals’ ad libitum although the feed was fed by hand each meal until the apparent satiety is achieved. The amount of feed intake at each meal is not fixed. This practice will ensure the fish eat what they want to eat and can determine the true differences in feed utilization and quality of different diets.

  1. Was this amount sufficient to feed the 20 fish in each aquarium? How did you determine this amount?
    Response:The answer is yes. In a feeding trial, an appropriate stocking density of tested fish is of importance for successful growth trials. The amount of fish used in this study can meet the stocking density requirements for the feeding trial and the fish fed the FM diets grew normally as expected based on our long-term feeding practice.

  1. What is the reason for choosing 56 days for the experiment period? How is this period determined?

Response: The experiment lasted for 56 days because fish can adapt to the experimental feed, and can reflect the results of growth performance, biochemical indicators and gene expression differences.

  1. (Page 5) About FI- There is no data about them (FI) in the text. What are these values? These values should be added to the result section.

Response: This is due to our negligence. Thank you for your kind reminder. We changed FI to FR in the text.

  1. Is there a special reason for choosing the liver for the study?

Response: There is no special reason for choosing liver for the study. Just because of its metabolic importance for nutrients as one of the most important metabolic organs in fish.

  1. Other suggestions are in the attached file

(1) If you did not set these values yourself, you need to add a reference (line 224-225, line 251-256).

Response: Thanks for your valuable suggestion, we added the reference.

(2)Why isn't a statement about IBW and FBW included in this section? If these are not specified, the reason was determined in the study (Table 3).

Response: Thanks to your kind reminding. We added the describe of IBW and FBW in the text and the notes below the Table 3.

(3)How do you interpret this result? (Table 3)

Response: Thanks for your careful checks. We are sorry for our carelessness. We corrected “abc” into “bc” in the Table 3.

Reviewer 2 Report

The ms is very well written, results are sound, the experimental and methodological approaches are appropiate and results are well discussed. Minor comments indicated in the pdf.

Author Response

Reviewer 2

The ms is very well written, results are sound, the experimental and methodological approaches are appropriate and results are well discussed. Minor comments indicated in the pdf.

  1. The proper term for it is: "dysbiosis" (line 14).

Response: We appreciate the reviewer’s suggestion and changed the expression.

  1. This is not enough for mentioning "destroy microbiota" as mentioned in the simple summary (line 35-36)

Response: Thanks for your valuable suggestion. We made a supplement in the simple summary.

  1. rephrase this term (line 40)

Response: Thanks to the reviewer's reminding. We replaced “intestinal microbiota balance” with “ intestinal microbiota profile”.

  1. provide name of manufacture and proximate composition of commercial feed (line 140).

Response: Thank you for your suggestion. The name of manufacture and proximate composition of commercial feed were added in the section 2.3.

provide information about amplicon size and efficiency of amplification (Table 2).

Response: As suggested by the reviewer, we added amplicon size and efficiency of amplification.

  1. Provide higher magnification with details ofinflammation in a single vill magnification bars are not readable in each figure (Figure 1)

Response: We sincerely appreciate your valuable comments. In order to observe the annotations more comprehensively and intuitively, we have adjusted the resolution of the whole Figure 1.

  1. appoptosis is visible in red cells. I do not see them, please indicate them with an arrow (Figure 4)

Response: As suggested by the reviewer, the apoptotic cells are marked with red arrows.

  1. I recommend moving this section before the microbiome analysis (line 471-491)

Response: We sincerely appreciate the valuable comments. We moved this section before the microbiome analysis and adjusted the order of the corresponding paragraphs and figures.

Reviewer 3 Report

In this study, the authors investigate the effects of dietary glycinin levels on the growth, intestinal barrier function, intestinal cell apoptosis and intestinal microbiotia in the intestine of orange-spotted grouper in order to further understand the relationship between intestinal health and SAPs in the fish species and clarify its mechanism of action, which provides a clue for the rational development and utilization of soybean by-products in grouper feed. However, I still cannot recommend publication with current form of their manuscript for several reasons.

Abstract

I consider that the abstract is too long. It includes practically all the results, with an excess use of acronyms. The abstract should be simpler, stating the problem, providing basic details of the experimental set-up and highlighting the most important results. Such as line 17-18 could be removed.

Introduction

Line 56. The major source alternative to FM is soybean by-products? That doesn't feel right. It's too absolute. At present, there are many studies on fish meal replacement, and some animal protein sources have better replacement effects than soybean meal, but soybean meal is cheaper than animal protein sources, but the cost performance may be higher.

Line 56-81. This part of the preface needs to be further improved, there are many studies on the negative effects of replacing fish meal with soybean meal, and there is no complete summary here. In their current manuscript, the authors should provide full backgroup information to describe why they want to carry out the experiment.

with the largest aquaculture production scale in China? There is ambiguity in this sentence.

Why was essential amino acid balance not included in the experimental design?

It is strongly recommended that authors present in Table 1, the chemical composition of the feed. Knowing levels of moisture, fiber and energy are essential for a correct evaluation of the study and formulation of experimental diets.

Before the start of the trial, the fish were fed with commercial feed for 4-week acclimatization. Where does commercial material come from? What are the nutrient levels?

Line 140. Add specific specifications for experimental fish.

the water temperature fluctuated between 28℃. Add the specific scope.

The authors include the standard error (SEM) when the standard deviation (SD) should be used, but also because they are determined with very few animals.

Moderate editing of English language required

Author Response

Reviewer 3

In this study, the authors investigate the effects of dietary glycinin levels on the growth, intestinal barrier function, intestinal cell apoptosis and intestinal microbiotia in the intestine of orange-spotted grouper in order to further understand the relationship between intestinal health and SAPs in the fish species and clarify its mechanism of action, which provides a clue for the rational development and utilization of soybean by-products in grouper feed. However, I still cannot recommend publication with current form of their manuscript for several reasons.

Abstract

  1. I consider that the abstract is too long. It includes practically all the results, with an excess use of acronyms. The abstract should be simpler, stating the problem, providing basic details of the experimental set-up and highlighting the most important results. Such as line 17-18 could be removed.

Response: We sincerely appreciate the valuable comments. We have made efforts to revise the abstract according to the suggestions by the reviewer.

Introduction

  1. Line 56. The major source alternative to FM is soybean by-products? That doesn't feel right. It's too absolute. At present, there are many studies on fish meal replacement, and some animal protein sources have better replacement effects than soybean meal, but soybean meal is cheaper than animal protein sources, but the cost performance may be higher.

Response: We agree with your comments. We rewrote the description as suggested.

  1. Line 56-81. This part of the preface needs to be further improved, there are many studies on the negative effects of replacing fish meal with soybean meal, and there is no complete summary here. In their current manuscript, the authors should provide full background information to describe why they want to carry out the experimentwith the largest aquaculture production scale in China? There is ambiguity in this sentence.

Response: Thanks to your valuable comments. We have adjusted the descriptions for this part.

4.Why was essential amino acid balance not included in the experimental design?

It is strongly recommended that authors present in Table 1, the chemical composition of the feed. Knowing levels of moisture, fiber and energy are essential for a correct evaluation of the study and formulation of experimental diets.

Response: We are not sure if our answer satisfies you, but we will try our best to answer your question and hope the answer meets your requirements. Our aim of the study is to explore the effects of soy antigen proteins glycinin on growth and biological functions of groupers. When designing experimental feed, it is generally necessary to first consider preparation of isonitrogenous, isolipidic, and/or isocaloric feeds. Based upon the practice of feed preparation, other nutritional factors can be considered to be tested. For example, utilization differences of feed ingredients. Due to the relatively low utilization of crystalline amino acids for most fishes including grouper, the EAA addition in the feed does not necessarily achieve ideal growth effects in the case of the feed with a balanced EAA profile. In addition, dietary addition of EAA may alter the utilization of EAA from intact protein, which covers the true effect of protein in feed ingredients. In the feed design, the crude protein, crude fat, and gross energy of high SBM diets are generally equivalent to those of the other three diets, although the high SBM diets contains less balanced AA profile than the other diets due to the high replacement proportion of fish meal by SBM. Besides fish meal, casein and gelatin are protein ingredients with relatively high quality protein. They are usually used in the research on the nutrition and feed utilization of fish. In order to ensure the good nutritional value of casein and gelatin when they are used in the feed formulations, the ratio of casein to gelatin in the feeds is 4:1, which is an appropriate proportion to realize a better AA balance and good utilization of the compound feeds. As suggested by the reviewer, we supplemented the gross energy values of the experimental diets.

5.Before the start of the trial, the fish were fed with commercial feed for 4-week acclimatization. Where does commercial material come from? What are the nutrient levels?

Response: Thank you for your suggestion. The name of manufacture and proximate composition of commercial feed were added in the section 2.3.

6.Line 140. Add specific specifications for experimental fish.

Response: We supplemented the name of fish species for experimental fish and added the specific specifications of experimental fish according to the reviewer’s suggestion.

7.the water temperature fluctuated between 28℃. Add the specific scope.

Response: As suggested by the reviewer, we added this in the text.

8.The authors include the standard error (SEM) when the standard deviation (SD) should be used, but also because they are determined with very few animals.

Response: Thank you for your kind suggestions. We checked all the results and changed all the SEM to SD in the text.

Round 2

Reviewer 1 Report

After the examinations, it was determined that the questions were answered appropriately and the suggested corrections were made.

Reviewer 3 Report

The MS has been well revised, has been greatly improved, agree to accept